# Personalized Lifestyle Interventions for Prevention and Treatment of Obesity-Related Cancers: A Call to Action

**DOI:** 10.3390/cancers17081255

**Published:** 2025-04-08

**Authors:** Mohamad Motevalli, Fatima Cody Stanford

**Affiliations:** 1Department of Sport Science, University of Innsbruck, 6020 Innsbruck, Austria; 2Harvard Medical School, Boston, MA 02115, USA; fstanford@mgh.harvard.edu; 3MGH Weight Center, Massachusetts General Hospital, Boston, MA 02114, USA; 4Nutrition Obesity Research Center at Harvard (NORCH), Massachusetts General Hospital, Boston, MA 02114, USA; 5Department of Pediatrics, Division of Endocrinology, Massachusetts General Hospital, Boston, MA 02114, USA; 6Department of Medicine, Division of Endocrinology-Neuroendocrine, Massachusetts General Hospital, Boston, MA 02114, USA

**Keywords:** oncology, carcinogenesis, tumorigenesis, health behavior, lifestyle medicine, precision medicine, overweight, prevention, health policy, public health

## Abstract

Cancer causes nearly 10 million deaths annually worldwide, with obesity being a major risk factor that directly contributes to many types of cancer. The increasing prevalence of both cancer and obesity necessitates improving and modernizing strategies for prevention and treatment, particularly for obesity-related cancers. Lifestyle modifications are crucial in cancer prevention and treatment; however, existing guidelines are still broad and lack the details needed for effective implementation. This article calls for collaboration across the research, healthcare, and policy sectors to develop personalized lifestyle interventions that consider genetic, demographic, physiological, environmental, and behavioral factors. Acting now can speed up the creation of a future where cancer prevention and treatment are tailored to each person’s needs.

## 1. Introduction

Cancer is a leading cause of premature death worldwide, with its incidence continuing to rise [1,2]. The International Agency for Research on Cancer (IARC) reports that cancer causes nearly 10 million deaths each year [3]. The global cancer burden is anticipated to reach 28.4 million cases by 2040, reflecting a 47% rise from 2020 [2]. While advancements in medical therapies such as immunotherapy and targeted treatments have significantly improved survival rates [4,5], there is increasing recognition that preventive approaches remain a powerful and cost-effective strategy for reducing the cancer burden [6,7].

Among the numerous risk factors associated with cancer, obesity has emerged as a critical determinant, particularly concerning cancers such as post-menopausal breast, colorectal, endometrial, esophageal, kidney, liver, pancreatic, and gallbladder [8,9], which are categorized as obesity-related cancers. These cancers are often accompanied by comorbid conditions such as diabetes, hypertension, and cardiovascular disease [10], which further complicate treatment and recovery, ultimately increasing the economic burden on healthcare systems.

While public health guidelines recommend lifestyle interventions, including dietary modifications, increased physical activity (PA), smoking cessation, and alcohol reduction, as strategies to reduce cancer risk [11,12], these recommendations often follow a one-size-fits-all approach. Research shows that the effectiveness of standardized lifestyle interventions varies greatly among individuals due to differences in genetics, metabolism, environment, and behavior [13,14]. For example, individuals with insulin resistance may respond differently to dietary interventions compared with those with normal glucose metabolism [15], and the same physical exercise program may result in varying metabolic effects depending on an individual’s genetic background [16] and/or body composition [17]. This highlights the need for applying personalized approaches to lifestyle modification, specifically in the context of obesity-related cancers.

Given the limitations of generic lifestyle recommendations in cancer prevention and management, there is a critical need to shift toward integrative, personalized approaches. This perspective article aims to advocate for the integration of personalized lifestyle interventions that account for demographic, genetic, physiological, environmental, and behavioral factors. It emphasizes the potential benefits of a comprehensive, tailored approach to improving patient adherence, enhancing therapeutic effectiveness, optimizing clinical outcomes, and ultimately mitigating the burden of obesity-related cancers.

## 2. Cancer and Obesity

Obesity has become one of the major global public health crises, with its prevalence nearly tripling over the past 50 years [18]. According to the World Health Organization (WHO), at least 59% of adults worldwide are currently living with excess body weight (overweight or obesity), and this statistic continues to rise [18]. However, the recent Lancet Commission definition and criteria of clinical obesity suggest that this previous estimate may be underestimated. The revised definition focuses on excess adiposity, highlighting that traditional measures based on body mass index (BMI) can fail to capture the true prevalence of obesity, potentially overlooking individuals who may be at risk of significant health complications due to organ and tissue dysfunction associated with excess adiposity [19].

The relationship between obesity and cancer has been recognized and studied in the scientific literature since the early 1900s [20]. Based on multiple meta-analyses, the IARC published a report identifying at least 13 types of cancer with a confirmed causal link to obesity, including breast, colorectal, endometrial, esophageal, gallbladder, kidney, liver, ovarian, multiple myeloma, meningioma, pancreatic, gastric, and thyroid cancers [21]. Particularly, excess body weight is associated with a 1.1 to 7 times increased risk of various cancers, with the highest risks observed for endometrial cancer, esophageal adenocarcinoma, and gastric cardia cancer [21,22]. This association often originates early in life, as research indicates that body fatness in youth is linked to an increased risk of developing eight types of cancer in adulthood [23]. Consistent data show that excess body weight is responsible for 14% of cancer deaths in men and 20% in women [24]. A recent meta-analysis found that a ~5 kg/m^2^ increase in BMI was associated with a higher risk of obesity-related cancer, with an 11% increase in individuals without cardiometabolic diseases and an 11–17% increase in those with type 2 diabetes or cardiovascular disease [25]. On the other hand, data show that intentional weight reduction is associated with a 12% [26] and 14% [27] reduced risk of obesity-related cancers. Obesity has also been linked to poorer cancer prognosis, with higher recurrence rates and reduced survival across multiple cancer types [22,28]. Results from a large-scale meta-analysis [29] indicate that obesity is associated with increased cancer mortality in individuals with breast, colorectal, prostate, and pancreatic cancers, as well as higher relapse rates in breast, colorectal, prostate, and gastroesophageal cancers. Independent of cancer-specific mortality, the meta-analysis also identified higher overall mortality among patients with breast, colorectal, or uterine cancers who were affected by obesity compared with those without. Additionally, the study found that obesity was linked to a 14% reduction in overall survival, a 17% decrease in cancer-specific survival, and a 13% increase in the risk of recurrence among cancer patients, while also highlighting that individuals with obesity show better survival outcomes in certain cancers, including lung cancer, renal cell carcinoma, and melanoma, compared with their non-obese counterparts [29].

The biological mechanisms underlying the obesity–cancer connection are complex and multifactorial, involving metabolic dysregulation, chronic inflammation, hormonal imbalances, and immune system alterations [9,30]. These pathways collectively create a pro-tumorigenic environment that promotes carcinogenesis, tumor progression, and resistance to therapy [30]. However, in certain cancers, including lung adenocarcinoma, endometrial carcinoma, and cancers of unknown primaries, obesity drives etiological heterogeneity by influencing specific driver mutations independent of clinical, demographic, and genetic factors [31].

Metabolic dysregulation, including insulin resistance and hyperinsulinemia, creates a pro-tumorigenic environment by increasing free insulin-like growth factor 1 (IGF-1) levels and enhancing hormone availability, which supports hormone-dependent cancers [32,33]. For instance, data show that a 5 nmol/L increase in IGF-1 levels is associated with a 10%, 9%, and 22% increased risk of breast, prostate, and thyroid cancers, respectively [34]. Chronic inflammation, marked by elevated pro-inflammatory cytokines and oxidative stress, induces DNA damage and tumor cell proliferation [35,36,37], with research indicating that approximately 20% of human cancers are associated with chronic inflammatory conditions [38]. Hormonal imbalances such as increased estrogen and leptin, along with decreased adiponectin, contribute to tumor growth [39,40], while immune system alterations, including macrophage polarization, T-cell dysfunction, and accumulation of myeloid-derived suppressor cells (MDSCs), further drive tumorigenesis [41,42].

Additionally, obesity-induced changes in the gut microbiota exacerbate inflammation and insulin resistance while increasing the production of carcinogenic metabolites, enhancing cancer risk, particularly for colorectal and liver cancers [43,44]. Research shows that 11 types of gut microbiota are linked to an increased risk of hepatocellular carcinoma and biliary tract cancer, establishing a causal connection and highlighting the gut–liver relationship in cancer development [45]. Furthermore, data indicate that 23–40% of identified gut bacterial species are associated with colorectal cancer or related health issues [46].

Moreover, appetite regulation and fat distribution differences can influence a patient’s response to lifestyle interventions. Notably, genetic predispositions that affect fat storage may lead some patients to accumulate excess fat in visceral tissues, which is more strongly associated with cancer risk than subcutaneous fat [47]. This correlation is also evident regarding central obesity, as data indicate that the risk of pancreatic cancer is 70% higher for individuals in the highest quintile of waist-to-hip ratio compared with those in the lowest quintile [48].

These biological mechanisms interact synergistically, creating a feedback loop that accelerates cancer development in individuals with obesity. Given the complex interplay of these factors, addressing obesity through personalized lifestyle interventions offers a promising approach for both cancer prevention and treatment. Understanding the role of obesity in cancer progression sets the stage for exploring how targeted lifestyle modifications can mitigate cancer risk and improve outcomes.

## 3. Lifestyle Modifications and Cancer

Research on monozygotic twins [49] shows that less than 10% of all cancers originate from inherited genetic defects. Consistent evidence indicates that most cancers are not inherited and that modifiable lifestyle factors and environmental conditions play a significant role in their onset [50,51]. Lifestyle modifications have long been recognized as essential for cancer prevention and management, with robust epidemiological and clinical evidence demonstrating that lifestyle interventions can significantly reduce the likelihood of cancer incidence and improve health outcomes in cancer patients. Key strategies for cancer prevention include avoiding tobacco, the leading cause of cancers like lung cancer, and adopting a nutritious diet rich in fruits, vegetables, and whole grains while minimizing processed and sugary foods to reduce inflammation [11,12,52,53,54,55]. An umbrella review of 19 studies across 22 cancer sites found that regular PA plays a significant role in reducing the risk of 7 cancers: breast, colorectal, endometrial, lung, esophageal, pancreatic, and meningioma [56]. As prominent examples, regular moderate-to-vigorous PA aids weight management and lowers the risk of breast and endometrial cancers by regulating hormone levels [12,53,57], and it reduces the risk of colorectal cancer by decreasing inflammation, improving gut motility, enhancing immune function, and regulating metabolic processes [58,59]. Additional strategies, such as limiting alcohol intake, ensuring adequate sleep, managing stress, and maintaining a healthy weight, further promote cancer prevention by enhancing overall health and reducing inflammation [12,52,54]. In this context, evidence shows that psychological distress and sleep quality are two significant factors in the prevention and progression of cancer [60,61]. Strategies like cognitive behavioral therapy, psycho-educational programs, and mind–body practices (such as yoga) have been well documented for their potential to manage stress and improve sleep quality in cancer patients, ultimately reducing cancer-related fatigue and improving quality of life [62,63].

On the other hand, data show that unhealthy lifestyle behaviors, including smoking, physical inactivity, and poor diet, are significantly associated with an increased risk of cancer [64,65]. Results from a large-scale study found that adherence to unhealthy lifestyle behaviors is associated with approximately a 1.5 times higher risk of all cancers [66]. Another study reports that sedentary behavior alone is associated with a 13% increase in all-cancer mortality [67], highlighting the significant impact of physical inactivity on overall cancer death rates.

Emerging practical trends have demonstrated lifestyle medicine programs’ effectiveness in enhancing cancer patients’ health outcomes. For example, multidisciplinary lifestyle medicine clinics supporting cancer survivors have shown promise in improving quality of life and overall health after treatment [68]. The Kaiser Permanente East Bay Lifestyle Medicine Program for breast cancer survivors demonstrated high patient satisfaction. It showed promise in increasing confidence in lifestyle changes while effectively addressing key concerns, such as sexual health and anti-hormonal therapy side effects, highlighting its potential for broader integration into oncology care [69]. However, a recent systematic review found that while associations between post-diagnosis lifestyle factors and health outcomes exist, the limitations of existing studies, including potential biases and confounding factors, contribute to low-quality evidence [53]. This is likely due to poor adherence to lifestyle medicine recommendations among cancer patients coupled with significant demographic variations [70]. While lifestyle modifications offer significant benefits in cancer prevention and treatment, a more personalized approach that considers individual differences is crucial for optimizing outcomes.

## 4. Personalized Lifestyle Approaches in Cancer Prevention and Treatment

Recent advances in systems biology and precision medicine highlight the importance of personalized approaches in every aspect of medicine, including cancer prevention and treatment. Precision medicine aims to tailor medical approaches to the unique biological characteristics of each individual, going beyond general therapies [71,72]. The complexity of obesity’s etiology and its role in carcinogenesis further emphasize the untapped potential for practical lifestyle approaches in addressing obesity-related cancers [73]. The European Prospective Investigation into Cancer and Nutrition (EPIC) explored how dietary patterns, PA, and weight loss affect cancer risk across populations [74]. The findings reveal that while lifestyle modifications are generally beneficial, their effectiveness may significantly vary depending on individual factors such as baseline metabolic health, hormonal status, and genetic variations [74].

While lifestyle approaches have shown broad efficacy at the population level in cancer management [11,12,52,53,54,55], their one-size-fits-all nature often overlooks individual variability, resulting in inconsistent outcomes across different populations. For example, international guidelines from the World Cancer Research Fund (WCRF) and the American Institute for Cancer Research (AICR) emphasize dietary patterns for cancer prevention, recommending increased intake of whole plant-based foods rich in micronutrients and antioxidants while limiting red and processed meats, refined carbohydrates, and added sugars [75,76]. However, these guidelines do not fully consider individual differences. For example, individuals with insulin resistance may require a lower carbohydrate intake [77,78], while cancer patients undergoing treatment, particularly for colorectal cancer, may experience altered nutrient absorption and need increased protein intake to support tissue repair and prevent muscle wasting [79], both of which may not align with general dietary recommendations. In addition, nutrigenomic research has identified specific dietary biocompounds, such as vitamin D and resveratrol, that may have a positive effect on colorectal cancer patients, highlighting these bioactive compounds as potential co-adjuvants in the personalized treatment of colorectal cancer [80]. Another example can be found in breast cancer research [81], where a study on patients with hormone receptor-positive (HR+) breast cancer undergoing endocrine therapy reported that a personalized postprandial glucose-targeting diet was more effective in improving weight management and glycemic control than the Mediterranean diet, a well-documented dietary approach for breast cancer survivorship [82].

Regarding alcohol, it has been well documented that excessive consumption is a well-established carcinogen, increasing the risk of liver, breast, esophageal, and colorectal cancers [83]. While current guidelines recommend limiting alcohol to no more than one drink per day for women and two for men [84], data show that individual responses to alcohol can vary due to genetic polymorphisms in alcohol metabolism enzymes, such as alcohol dehydrogenase 1B (ADH1B) and aldehyde dehydrogenase 2 (ALDH2) [85], suggesting that personalized alcohol risk assessments could help refine and tailor these recommendations to better account for genetic differences. Although genetic testing for these polymorphisms is not yet widely available for the general population, advancements in genetic screening and precision medicine may allow for more tailored recommendations in the future. At the population level, however, targeted public health messaging may already be feasible. For example, East Asians may be more susceptible to the carcinogenic effects of alcohol due to the high prevalence of the ALDH2*2 allele in this population [86], which reduces ALDH2 enzymatic activity by 60–90% [87], suggesting that alcohol consumption guidelines could be adjusted to account for genetic susceptibility in certain demographic groups.

Regarding PA, a key component of cancer prevention and health promotion, the American Cancer Society (ACS) and the WHO recommend at least 150 min of moderate- to vigorous-intensity exercise per week and strength training at least twice a week [88,89]. While the evidence demonstrates the potential of these recommendations in reducing the risk of various cancers, improving immune function, and enhancing the quality of life for cancer survivors [90,91], it is crucial to recognize that the efficacy of exercise interventions can vary significantly due to various factors. For example, individual differences in physical limitations, such as joint pain or chronic conditions like arthritis, may reduce the effectiveness of certain exercise routines in achieving optimal outcomes [92,93]. Similarly, variations in baseline fitness levels can affect the effectiveness of exercise interventions, as individuals with lower fitness may require modified programs to achieve the same benefits as those with higher fitness levels [94,95]. While cancer patients undergoing treatment (e.g., chemotherapy, radiation) may require modified exercise programs to account for fatigue, sarcopenia, or immune suppression [95,96], the timing of exercise relative to treatment cycles, the severity of side effects, and the presence of comorbidities can further complicate the design and efficacy of exercise interventions. Therefore, tailoring exercise programs to these individual circumstances is essential to maximizing the therapeutic benefits of physical exercise while minimizing potential risks.

Different types of cancer have diverse biological mechanisms, and personalized lifestyle interventions should be strategically designed to target their specific mechanisms. Each cancer type and its associated biological pathways necessitate particular interventions tailored to the unique pathophysiological characteristics of the disease. For example, interventions should focus on reducing chronic inflammation in inflammation-driven cancers such as colorectal cancer. Diets rich in anti-inflammatory foods (such as omega-3 fatty acids, antioxidants, and fiber) and regular moderate-intensity aerobic exercise can help lower inflammatory markers and reduce cancer risk [91,97]. These strategies inhibit pro-inflammatory cytokines and enhance immune function, preventing cellular damage. For hormone-driven cancers like postmenopausal breast cancer, reducing excess body fat is crucial, as adipose tissue increases estrogen levels and drives cancer progression. Personalized interventions should include weight loss, phytoestrogens (e.g., soy), and physical exercise to regulate estrogen levels and reduce the hormonal drive for cancer [98,99,100]. In addition to cancer type, the stage of cancer can also influence tailored lifestyle interventions. In the early stages, the focus is on preventing progression and recurrence through improving physiological and mental health, strengthening the immune system, and supporting treatment responses. In advanced stages, when cancer has spread or become more aggressive, interventions focus on addressing malnutrition, managing treatment side effects, and reducing fatigue, thus making personalized supplementary strategies crucial alongside standard treatment routines. Overall, tailoring interventions to the specific cancer type, underlying biological mechanisms, and timing of implementation enhances both prevention and treatment efficacy.

Over the past decade, advancements in predictive modeling and digital health technologies have helped advance personalized lifestyle interventions, particularly cancer prevention and management. These innovations aim to enhance accessibility, precision, and scalability in personalized care. The Moving On study demonstrated the significant potential of mHealth interventions in supporting behavior change among cancer survivors [101]. Predictive models such as the Individual Survival Distribution (ISD) model utilize lifestyle and health history data to estimate cancer onset probabilities and guide actionable lifestyle changes [102]. Humanistic Health Management (HHM) also emphasizes personalized interventions tailored to lifestyle habits, psychological well-being, and health conditions, showing promising results [103]. However, the public perception of personalized cancer risk information varied. While some individuals found risk assessments motivating, those who perceived their risk as low were less inclined to change their behavior [104], highlighting the need for effective communication strategies to encourage proactive health decisions.

A comprehensive approach to cancer management should consider the complex interplay of biological, lifestyle, and environmental factors in disease progression. Preventive and therapeutic strategies in cancer management should be customized to an individual’s genetic and physiological profile, demographic features, behavioral characteristics, and environmental factors. For example, genetic makeup, such as inherited conditions or mutations, plays a critical role in determining cancer risk and guiding treatment options [105]. In addition, understanding how the body responds to treatments, such as how it metabolizes certain medications (pharmacogenomics), can help optimize drug dosing and selection [106]. Regarding demographic factors (such as age, sex, and socioeconomic status), evidence indicates that older adults, for instance, may need treatment adjustments to account for age-related health concerns [107]. Socioeconomic status may affect healthcare access, adherence to preventive measures, and treatment compliance [108], which may require personalized interventions such as financial support or more flexible treatment schedules. Environmental factors, such as exposure to pollution or workplace hazards, can further guide tailored strategies. For example, a person living in an area with high levels of air pollution may need to take steps to reduce exposure, like using air purifiers or moving to a less polluted environment [109]. In cases where an individual works in a hazardous occupation, regular screening and protective measures may be essential.

The essence of personalized cancer management is therefore a holistic approach that addresses all factors related to the etiology, development, and progression of the disease. Focusing on only a few aspects may overlook broader interactions, potentially limiting intervention effectiveness. This approach emphasizes that the mechanisms driving cancer progression vary not only between cancer types but also among patients with the same cancer, with a greater magnitude in individuals with obesity. Figure 1 presents a conceptual model highlighting the complex and multifaceted interactions between the factors affecting cancer management across five categories: genetics, behavior, environment, demographics, and physiology.

## 5. A Call to Action

Despite strong evidence supporting the role of lifestyle in cancer prevention, its integration into effective clinical practice remains limited. The future of cancer prevention and treatment, particularly for obesity-related cancers, relies on an integrative, data-driven, and practical approach. This includes individualized lifestyle modifications, digital health technologies, interdisciplinary collaboration, and policy-level initiatives. To advance this vision, researchers, clinicians, and policymakers should prioritize and accelerate the development, validation, and implementation of personalized lifestyle medicine, ensuring its seamless adoption within mainstream oncology and public health frameworks.

### 5.1. Researchers and Scientists

To advance the science of personalized lifestyle medicine, researchers must prioritize several critical areas that will drive the future of cancer prevention and treatment. One of the foremost priorities is identifying and validating biomarkers that predict cancer risk, treatment response, and the effectiveness of lifestyle interventions. Key research directions include exploring genomic and epigenetic markers that influence metabolic dysfunction and cancer susceptibility, metabolomic profiling to tailor dietary and exercise recommendations, and investigating gut microbiome signatures that mediate obesity–cancer interactions and affect treatment outcomes. Additionally, hormonal and inflammatory markers could guide precision nutrition and exercise prescriptions, offering a more individualized approach to cancer care. Researchers must also shift from traditional intervention trials to more personalized clinical trial designs that account for individual variability. This transition may involve stratified randomized controlled trials (RCTs) based on demographic, genetic, physiological, environmental, or behavioral data, adaptive trials that use real-time biomarker feedback to refine interventions, and N-of-1 trials that allow participants to test and adjust personalized interventions.

The rapid advancement of digital health technologies offers a significant opportunity to scale personalized interventions [110]. Researchers should prioritize evaluating the effectiveness of AI-driven health coaching, wearable sensors, mHealth and eHealth programs, and developing predictive AI models that integrate individualized data to create tailored cancer prevention strategies. Investigating tools like continuous glucose monitoring, digital phenotyping, and remote patient monitoring will be essential for advancing personalized lifestyle medicine. Lastly, to ensure equitable access to these innovations, research must address health disparities through community-based participatory research and explore the socioeconomic and cultural barriers to adopting precision lifestyle interventions.

### 5.2. Clinicians and Healthcare Providers in Cancer Care

To successfully integrate precision lifestyle medicine into cancer care, clinicians and healthcare providers must have the knowledge and tools to incorporate personalized data into patient treatment plans. Key initiatives to support this include developing educational programs on nutrigenomics, precision exercise medicine, and metabolic profiling alongside establishing clinical guidelines for incorporating biomarker-driven lifestyle interventions into oncology practice. Expanding continuing medical education programs focused on AI-driven decision support tools and digital health applications is also essential. Healthcare providers should leverage multi-omics data to customize dietary, physical activity, and behavioral interventions while integrating continuous monitoring tools such as wearable devices and sensors into patient management. Collaboration with interdisciplinary teams, including nutritionists, exercise physiologists, and behavioral health experts, is crucial for providing comprehensive, patient-centered care.

Additionally, remote monitoring and telehealth solutions should be promoted to ensure long-term patient engagement, adherence to interventions, and effective secondary prevention. Expanding precision lifestyle medicine clinics within oncology centers, offering personalized nutrition and exercise counseling, is another critical step, alongside digital twin technology, to predict individual responses to interventions. Lastly, the development of personalized survivorship plans for cancer survivors, particularly those at risk of recurrence, should be prioritized to incorporate tailored lifestyle modifications, ultimately enhancing outcomes and preventing future cancer incidences.

### 5.3. Policy- and Decisionmakers in Public Health

Policymakers must take comprehensive action across several key areas to establish precision cancer prevention and public health initiatives as standard practice. Governments must ensure equitable access to precision lifestyle interventions by subsidizing tailored healthcare services for underserved populations, expanding insurance coverage for personalized treatments, and integrating biomarker-driven prevention strategies into national cancer control programs to reduce healthcare disparities. Access to healthy foods is one of the most fundamental factors in cancer risk. Today, a considerable portion of the underdeveloped world is classified as “food deserts”, lacking grocery stores and fresh produce, while a significant portion of urban areas worldwide are “food swamps”, overloaded with fast-food outlets [111]; data shows that both environments are linked to higher obesity-related cancer mortality rates [111]. Policymakers should implement strategies such as enacting zoning laws to control the concentration of fast-food outlets and offering tax incentives to fresh food providers and markets to reverse these detrimental food access trends and promote healthier, more sustainable food environments.

Expanding public health funding is crucial for advancing precision prevention strategies. Governments and funding agencies should prioritize research grants for biomarker-driven lifestyle interventions, invest in developing AI-powered health technologies to improve access to personalized treatments, and encourage public–private partnerships to drive innovation in precision nutrition and exercise medicine. In addition, regulatory frameworks must be established to ensure the safety, efficacy, and ethical use of digital health technologies in oncology and broader preventive healthcare applications. Public awareness and education must also be prioritized, with policymakers launching campaigns to inform the public about the importance of precision lifestyle in cancer prevention, implementing school-based health programs, and promoting corporate wellness initiatives that incorporate personalized interventions. These efforts are crucial to ensuring the widespread adoption of precision lifestyle medicine and tackling the growing burden of cancer through personalized, equitable, and sustainable strategies.

## 6. Conclusions

The increasing prevalence of both cancer and obesity demands a fundamental shift in prevention and treatment strategies, especially for obesity-related cancers. Personalized lifestyle interventions that consider an individual’s demographic, genetic, physiological, environmental, and behavioral factors offer a more effective and tailored strategy than traditional methods while also leveraging cutting-edge technology and interdisciplinary collaboration. An integrated precision management approach should tailor cancer prevention strategies and guide comprehensive treatment after diagnosis, accounting for the dynamic interplay of factors that influence disease onset, progression, and treatment response with variations across different cancer types and stages. To unlock the full potential of precision lifestyle medicine, fostering collaboration across the research, healthcare, and policy sectors is essential. Decisive action today can accelerate advancements toward a future where cancer prevention and treatment are precisely tailored to each individual’s unique health profile. This approach can improve patient outcomes, enhance quality of life, and strengthen global cancer control efforts. The time for action is now.

## Figures and Tables

**Figure 1 cancers-17-01255-f001:**
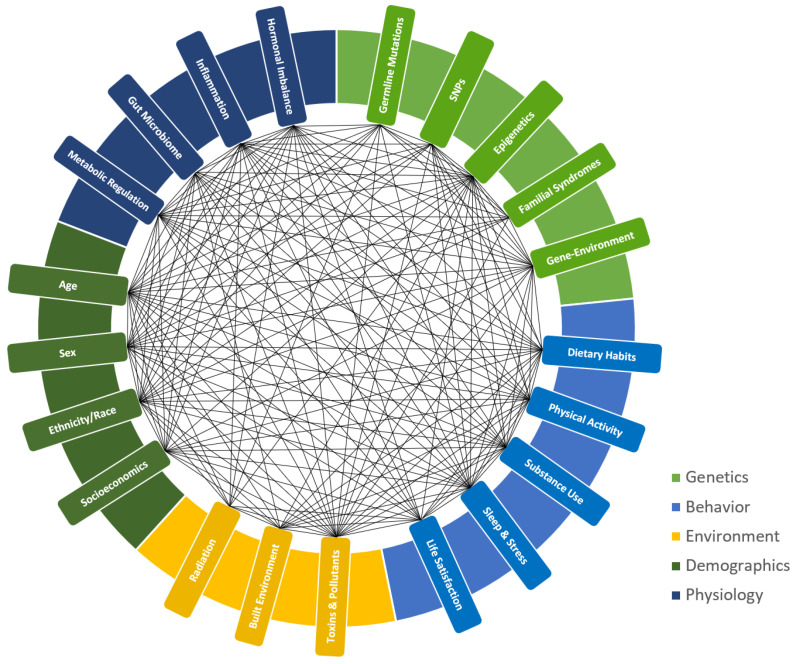
A conceptual model highlighting the complex and multifaceted interactions between key factors influencing cancer prevention and treatment. These factors are categorized into genetics, behavior, environment, demographics, and physiology. Genetic factors include germline mutations, single nucleotide polymorphisms (SNPs), epigenetics, familial syndromes, and gene–environment interactions. Physiological factors encompass hormonal imbalances, inflammation, the gut microbiome, and metabolic regulation. Demographic factors include age, sex, ethnicity/race, and socioeconomic status. Environmental factors involve exposure to toxins and pollutants, the built environment, and radiation. Behavioral factors include dietary habits, physical activity, substance use, sleep and stress, and overall life satisfaction.

## Data Availability

Not applicable.

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
