# Peer review of "Personalized Lifestyle Interventions for Prevention and Treatment of Obesity-Related Cancers: A Call to Action"

_cancers, 2025, doi:10.3390/cancers17081255_

Round 1

Reviewer 1 Report

Comments and Suggestions for Authors

The manuscript presents an insightful discussion on personalized lifestyle interventions for the prevention and treatment of obesity-related cancers. The authors provide a well-structured argument supported by relevant literature and highlight the importance of precision medicine in cancer prevention. The topic is timely, and the manuscript offers valuable contributions to the field of oncology and lifestyle medicine. However, there are several areas that require improvement in terms of clarity. 

The discussion on genetic and metabolic influences on obesity-related cancers is informative but requires more quantitative data to support claims. For example, several claims would benefit from additional references, particularly in the sections discussing behavioral interventions and obesity.

I have suggested including tables to summarize key findings,  and to improve transitions between sections.

While the manuscript presents a compelling case for personalized lifestyle interventions, improvements in methodological clarity, organization, and supporting evidence are necessary before publication.

Reviewer 2 Report

Comments and Suggestions for Authors

Obesity is still an insufficiently recognized risk factor for a several different cancer entities. This article discusses the role of obesity for the development of cancer and emphasizes personalized interventions. As stated in the title, the authors see their article as a wake-up call. It can be regarded as a position paper and does not follow the conventional structure of (narrative) reviews. Following an introduction with a general overview on cancer risk factors and interventions, there are chapters dealing with cancer and obesity, general lifestyle interventions, personalized lifestyle interventions in cancer prevention and treatment, followed by what the authors termed a call to action for researchers and scientists, clinicians and health care providers and policy- and decision makers, and final concluding remarks. One figure displaying a conceptual model of complex and multifaceted interactions between key factors is included.

The paper is well-written and readable. The main problem with this manuscript lies, however, in the particular format chosen by the authors. The approach attempts to put together a series of elements to call for interventions to reduce body weight, but cannot cover the various aspects in depth or completeness. Mostly presenting one or two examples for their statements and theses, respectively, the authors stop short from providing comprehensive information to the various topics they address. For the readers of Cancers, a journal dedicated to basic, translational, and clinical studies on all tumor types, some more specific information on the value of evidence-based interventions for this important goal would presumably be desirable. Thus, I add some suggestions that could render this work more concretely practical and helpful for the reader

For instance, the authors state correctly in lines 97-99 that obesity has also been linked to poorer cancer prognosis, with higher recurrence rates and reduced survival across multiple cancer types with two references (from 2013 and 2022). This might be highly interesting to clinical oncologists but unfortunately no further details are provided to this topic. More recent references, and an overview on the cancer entities, for which evidence is currently available, might be helpful, even more so if the respective level of evidence would be provided as well. This could be presented in a table.

Moreover, it is reported that obesity-induced changes in gut microbiota exacerbate inflammation and insulin resistance while increasing the production of carcinogenic metabolites, thereby enhancing cancer risk for colorectal and liver cancers (lines 113-115). This topic, too, would merit some further explanation instead of merely being mentioned in a sole sentence with two references.

The authors criticize the one-size-fits-all nature of approaches for lifestyle interventions and emphasize individual variability. As an example, they cite conflicting recommendations for colon cancer and diabetic patients. Again, more detailed examples and the available evidence, respectively, might help render this statement more practical for health care professionals. This could be done with a table of figure.

The authors also criticize recommendations limiting alcohol consumption to no more than one drink per day for women and two for men, referring to genetic polymorphisms in alcohol metabolism. They suggest that personalized alcohol risk assessments could help refine and tailor recommendations for alcohol to better account for genetic differences (lines 187-192). It appears, however, not conceivable how this might be accomplished since general genetic testing for polymorphisms of alcohol metabolism is unavailable for the general population.

With respect to the statement that different types of cancer have diverse biological mechanisms, and personalized life-style interventions should be strategically designed to target their specific mechanisms (lines 211-212), it appears difficult to present general recommendations for the public. Accordingly, in lines 247-253 it is stated that “a comprehensive approach to cancer management should consider the complex interplay of biological, lifestyle, and environmental factors in disease progression”, and “preventive and therapeutic strategies in cancer management should be customized to an individual’s genetic and physiological profile, demographic features, behavioral characteristics, and environmental factors”. While recommendations remain still vague, a table listing specific and practical ways to achieve such a goal might be helpful.

The above-mentioned complex interplay is depicted in Figure 1, showing “a conceptual model highlighting the complex and multifaceted interactions between key factors”. It is rewarding to point to the complex interactions of factors, however, the practical use of the information contained in this figure for designing personalized lifestyle interventions for prevention and treatment of obesity-associated cancers is somewhat limited since almost every item is related to all other items.

There are some inaccuracies in this papers that should be clarified or corrected. In lines 134-136, the authors write that physical activity “… lowers the risk of cancers such as breast and colon by regulating hormone levels“. While sex hormones are thought to play a role in the development of breast cancer, this may not be the case for colon cancer (for instance: Michels KA et al., Endogenous Progestogens and Colorectal Cancer Risk among Postmenopausal Women. Cancer Epidemiol Biomarkers Prev. 2021;30:1100-1105). Furthermore, a proposed connection between adequate sleep and stress management and cancer prevention (lines 137-138) is still speculative and lacks good clinical evidence.

Reviewer 3 Report

Comments and Suggestions for Authors

In this manuscript, Drs Motevalli and Stanford present their view that the prevention and treatment of cancer must be tailored to each person’s needs.  While this is generally the course that medical treatment is headed, they encourage more resources, time, and money to be dedicated to developing appropriate tools so clinicians can treat each patient based on their unique health profile. In their paper, they provide evidence that we have the foundation for these tools to be developed, but they are not being adequately supported to unlock the full potential of precision lifestyle medicine. Their point of view is well supported by peer-reviewed literature, and they present their case in a logical order. I don’t have any additional comments on the text. The authors present a conceptual model in Figure 1. They should consider adding access to foods that are good for our health and geographical location which has been linked to climate change and cancer risk.

Round 2

Reviewer 2 Report

Comments and Suggestions for Authors

The authors were responsive to most comments and have markedly improved their manuscript. Improvement is particularly noticeable in section, which is 3 dealing with lifestyle modifications.

Furthermore, the addition of citation 29 with the meta-analysis by Petrelli et al. (2021) is highly relevant for this subject. The authors have decided not to include a table with main findings, which is o.k.. Nonetheless, it would be very useful for readers who are not very familiar with this topic if the authors would delve deeper into the consistency and discrepancies of this meta-analysis (and maybe related papers). For instance, obese patients with certain cancers (e.g., uterine cancer) experience an increased risk not only for cancer-specific, but also for overall mortality, Of note, even if this contradicts the authors' intentions, it should not be completely concealed that in some tumor entities such as malignant melanoma, renal cell carcinoma or lung cancer, patients with obesity have better survival rates.

Minor point: The full name should be provided when an abbreviation is first used (e.g., HR+).

In summary, I recommend acceptance of this article for publication with some additions regarding the risk of developing certain cancers and  implications for prognosis after treatment. 
